# Reuse of Molecules for Glioblastoma Therapy

**DOI:** 10.3390/ph14020099

**Published:** 2021-01-28

**Authors:** Abigail Koehler, Aniruddha Karve, Pankaj Desai, Jack Arbiser, David R. Plas, Xiaoyang Qi, Renee D. Read, Atsuo T. Sasaki, Vaibhavkumar S. Gawali, Donatien K. Toukam, Debanjan Bhattacharya, Laura Kallay, Daniel A. Pomeranz Krummel, Soma Sengupta

**Affiliations:** 1Department of Neurology and Rehabilitation Medicine, University of Cincinnati College of Medicine, Cincinnati, OH 45267, USA; koehleai@ucmail.uc.edu (A.K.); gawalivr@ucmail.uc.edu (V.S.G.); kamdemde@ucmail.uc.edu (D.K.T.); bhattadj@ucmail.uc.edu (D.B.); kallaylm@ucmail.uc.edu (L.K.); krummedl@ucmail.uc.edu (D.A.P.K.); 2Division of Pharmaceutical Sciences, University of Cincinnati James L. Winkle College of Pharmacy, Cincinnati, OH 45229, USA; karveas@mail.uc.edu (A.K.); desaipb@ucmail.uc.edu (P.D.); 3Department of Dermatology, Emory School of Medicine, Atlanta, GA 30322, USA; jarbise@emory.edu; 4Atlanta Veterans Administration Medical Center, Decatur, GA 30033, USA; 5Department of Cancer Biology, University of Cincinnati College of Medicine, Cincinnati, OH 45267, USA; plasd@ucmail.uc.edu; 6Department of Internal Medicine, University of Cincinnati College of Medicine, Cincinnati, OH 45267, USA; qix@ucmail.uc.edu (X.Q.); sasakiao@ucmail.uc.edu (A.T.S.); 7Department of Pharmacology and Chemical Biology, Emory School of Medicine, Atlanta, GA 30322, USA; renee.read@emory.edu

**Keywords:** glioblastoma, brain cancer, letrozole, S6K1 inhibitors, imipramine blue, Visudyne^®^, CellCept^®^, saposin C

## Abstract

Glioblastoma multiforme (GBM) is a highly malignant primary brain tumor. The current standard of care for GBM is the Stupp protocol which includes surgical resection, followed by radiotherapy concomitant with the DNA alkylator temozolomide; however, survival under this treatment regimen is an abysmal 12–18 months. New and emerging treatments include the application of a physical device, non-invasive ‘tumor treating fields’ (TTFs), including its concomitant use with standard of care; and varied vaccines and immunotherapeutics being trialed. Some of these approaches have extended life by a few months over standard of care, but in some cases are only available for a minority of GBM patients. Extensive activity is also underway to repurpose and reposition therapeutics for GBM, either alone or in combination with the standard of care. In this review, we present select molecules that target different pathways and are at various stages of clinical translation as case studies to illustrate the rationale for their repurposing-repositioning and potential clinical use.

## 1. Introduction

### 1.1. Molecular Classification

Glial tumors can be divided into two categories: diffuse and circumscribed [1]. Diffuse tumors are highly likely to recur due to their nature of malignancy by infiltrating surrounding brain tissue, as opposed to the benign growth pattern of circumscribed tumors. Diffuse gliomas can further be categorized as WHO grades II, III, or IV tumors. Glioblastoma multiforme (GBM) is synonymous with a WHO grade IV malignancy and accounts for more than half of all adult primary brain tumors [1,2]. In adult populations, primary tumors are typically more likely to affect elderly patients, whereas secondary tumors typically affect patients 45 years of age or younger [2,3]. GBMs can be primary tumors, signifying they are grade IV at baseline or secondary tumors that have evolved from lower grade tumors. Low grade histology divisions include astrocytoma, oligodendroglioma, oligoastrocytoma, and the three aforementioned anaplastic forms [1,3]. The four major genetic and epigenetic irregularities noted in GBM are derived from mutations in the metabolic enzyme isocitrate dehydrogenase 1 and 2 genes (IDH1/2), amplification in the epidermal growth factor receptor (EGFR), amplification of platelet derived growth factor alpha (PDGFRA), and the loss or mutation of neurofibromatosis type 1 gene (NF1) [1,3,4,5]. Primary tumors often show a high level of gene expression or mutation in oncoproteins such as EGFR or NF1 loss or mutation, while secondary GBMs typically express mutations in IDH1/2 [1,3,4,5]. IDH wild type is most consistent in GBM primary tumors, whereas IDH mutant is consistent with low-grade gliomas and secondary GBM [4]. GBMs can be further divided into four subtypes based on genomic abnormalities. These four subtypes are proneural, neural, classical, and mesenchymal. Previous studies have shown that mesenchymal subtypes have lower NF1 expression, but more specifically, focal hemizygous deletions of a region at 17q11.2 which contains the gene NF1 [5]. Proneural subtypes are often associated with younger age patients [3]. They express alterations in the PDGFRA gene with either higher amplification of the locus at 4q12 or multiple point mutations, and they also express point mutations in IDH1 [5]. Higher levels of PDGFRA amplifications are most often seen in pediatric GBMs, although childhood GBM is less common [1]. The neural subtype is classified by expression of neuron markers including NEFL, GABRA1, SYT1, and SLC12A5 [5]. Neuron projection and axon and synaptic transmission are gene ontologies associated with this subtype [5]. The classical subtype is commonly characterized by EGFR amplification or mutation [5]. Knowledge of the genetic discrepancies, tumor origination, histology, and DNA methylation patterns allow for more precise identification of tumors which predicts patient prognosis and guides possible treatment options.

### 1.2. Cellular Pathways in GBMs

GBMs rely heavily on different cellular pathways for growth, signaling, proliferation, and migration, among other things. The receptor tyrosine kinase (RTK) pathway is a major pathway in which GBM malignancies capitalize. Receptors include EGFR, vascular endothelial growth factor receptor (VEGFR), PDGFR, hepatocyte growth factor receptor (HGFR/c-MET), fibroblast growth factor receptor (FGFR), and insulin-like growth factor 1 receptor (IGF-1R) [6]. When these receptors are bound with a ligand, they trigger two RTK pathways: Ras/MAPK/ERK and PI3K/ATK/mTORC [6]. In the Ras/MAPK/ERK pathway, the Ras protein is activated through phosphorylation of GDP to GTP [6]. Ras activation leads to MAP kinase activation which then activates ERK through phosphorylation [6]. Activation of this pathway promotes tumorigenesis, cell proliferation, cell migration, and angiogenesis through increased VEGF expression [6]. The PI3K/ATK/mTORC pathway is activated by transmembrane tyrosine kinase growth factor receptors and integrins, and G-protein-coupled receptors [6]. A series of events occur to activate ATK, mTORC, and S6K1 [6]. PTEN works to counteract the activation of PI3K signaling by dephosphorylating PIP_1_ and PIP_2_, which are directly responsible for activating ATK [6]. This pathway is also responsible for inhibiting p53 and I_K_B, which are known for anti-tumor progression [6]. The PI3K/ATK/mTORC pathway leads to GBM cell survival, growth, proliferation, and even angiogenesis due to increased VEGF expression [6]. This pathway is found to be altered in nearly 86–90% of GBM cases studied in a recent review [6]. 

### 1.3. Current Treatment Options

Despite advances in molecular studies and multimodal treatment approaches, the prognosis of GBM patients remains dismal [7], with a median survival of ~14 months [8]. Therefore, there is a critical demand for new, life-extending approaches. Upon diagnosis, GBM patients typically follow the current standard of care, known as the Stupp protocol, undergoing maximal safe tumor resection. This is most often followed by adjuvant radiation and chemotherapy. Temozolomide, a DNA alkylating agent approved more than two decades ago, remains the primary chemotherapeutic for newly diagnosed GBMs [9]. Unfortunately, recurrence is observed in almost all patients, with limited therapeutic options available thereafter [7,10]. Most often recurrent GBM patients receive bevacizumab (brand name: Avastin^®^), a monoclonal antibody, for palliative support. Other options for the newly diagnosed and recurrent treatment include application of an FDA approved physical device, non-invasive alternating electric field therapy or ‘tumor treating fields’ (TTFs), including its concomitant use with standard of care. TTFs, administered through use of the Optune^®^ device, are most commonly applied to supplement treatment therapies to halt tumor growth [11]. Vaccines and immunotherapy have shown a degree of effectiveness for prostate cancer and melanoma, albeit responses are not durable [12]. Trials are ongoing with both approaches for a subset of qualifying GBM patients. Vaccines offer a boost to a patient’s immune system, which may prompt a response to tumor antigens [12]. The intent is that vaccinations, following the completion of the standard of care, will initiate an immune response for tumor antigens in the event of recurrence. 

### 1.4. Barriers to Identifying Effective Treatment

Barriers to the development of new therapeutic agents for GBMs include: (1) lack of selective, novel “druggable” targets; (2) inability of most drugs to cross the blood-brain barrier (BBB), penetrate the brain-tumor barrier (BTB), and selectively accumulate in tumor cells [13]; (3) molecular heterogeneity of GBMs [14]. Regarding the BBB/BTB, dysfunctional BBB/BTB as well as abnormal blood vessels, stem from hypoxic environments caused by metabolic demands of gliomas which increase angiogenesis and VEGF expression [11]. Abnormal blood vessels allow oxygen and nutrient delivery to the tumor and enable cell migration [15]. It is also important to note that the majority of patients undergoing treatment for GBMs develop resistance to standard of care therapy [13]. 

### 1.5. Repurposing and Repositioning Drugs

To accelerate treatment for GBMs in a cost-effective manner, investigators have turned to repositioning and/or repurposing FDA approved therapeutics with properties likely to confer BBB permeability. Identifying drugs to repurpose can be achieved by in silico screening; for example, repurposing of the antifungal drug itraconazole as an anti-cancer agent [16] or molecular target screening using sequencing and proteomic analysis of the tumors to provide a rational, personalized treatment [17]. Alternatively, anti-cancer drugs are being repositioned as therapeutics for GBM; for example, employing CDK 4/6 inhibitors commonly used to treat breast cancers as anti-GBM therapeutics [18].

Repurposing of FDA approved therapeutics can often utilize the “505(b)(2)” new drug application (NDA) approval pathway. Unlike the standard 505(b)(1) NDA regulatory submission pathway for new chemical entities that require complete safety and effectiveness reports from studies conducted by sponsor, the 505(b)(2) regulatory pathway allows sponsors to include information from published studies and findings of safety and effectiveness from approved products with the same active ingredient even when the studies were not conducted by the sponsor. The 505(b)(2) regulatory submission can significantly reduce the time of NDA approval and reduce product development costs for repurposed approved FDA therapeutics. While repurposing can significantly reduce the time, cost, and risk of drug development, drug repurposing is not without financial, legal and regulatory pitfalls and challenges. FDA approved therapies are protected against competition by both patents and data exclusivity granted at the time of FDA approval, which enable companies to recover development costs for new medicines. Patent terms are set for 20 years and protect the product’s intellectual property while exclusivity restricts the use of data generated by the drug innovator and prohibits approval of generic versions for a set time. The exclusivity period is 5 years for new chemical entities, 7 years for orphan drugs, and an additional 3 years of exclusivity for new clinical investigation of a previously active agreement with 6 months added to both pediatric and exclusivity for pediatric development. Biologic products with often complex, costly, and lengthy development may be granted up to 12 years of exclusivity. For off-patent products, development of the novel indication must be assessed relative to competition from the available generic market. For products under patent protection or within the exclusivity period, licensing agreements or partnerships must be established with the innovator company for product development. Drug repurposing also faces challenges attracting funding and industry support without clear marketing opportunities. Undoubtedly, collaboration between industry and/or biotechnology and academia is important to provide pharmaceutical expertise and funding sources that meet patient, investor, and regulatory needs for successful drug repositioning [19,20,21]. 

In this review, we focus on repurposing and repositioning select small molecule drugs as GBM therapeutics in adult tumors. Elsewhere, recent reviews have surveyed repurposed drugs for treating GBMs [13,14]. We aim to illustrate the pathway from bench to bedside and thus serve as a guide for translation of other molecules into the GBM space.

## 2. Repositioning a Small Molecule Inhibitor Used in Breast Cancer That Targets Estrogen Biosynthesis

Aromatase (estrogen synthase) is a member of the cytochrome P450 superfamily of proteins residing in the endoplasmic reticulum. Aromatase catalyzes the conversion of estrogen from androgen by aromatization of the A-ring (Figure 1A). Importantly, aromatase transforms androstenedione to estrone and testosterone to estradiol. We have known for some time that endogenous estrogens play an important role(s) in female reproductive development. Aromatase inhibitors (AIs), such as exemestane (brand name: Aromasin^®^) and letrozole (brand name: Femara^®^), are drugs used to inhibit estrogen synthesis. Letrozole is a member of the nonsteroidal benzyltriazole aromatase inhibitors and consists of a triazole ring, serving as a major functional group (Figure 1B) [22]. Letrozole is the first line of therapy in postmenopausal, hormone responsive breast cancer patients, and has been reported to contribute to incremental improvements in survival, lower recurrence rates, and lower incidence of contralateral breast cancers [23]. Previously, aromatase has been reported to play a proliferative and neuroprotective role in brain tissue [24]. More recently, evidence has accumulated that endogenously produced steroid hormones, such as estrogens, have a role(s) in development of primary and metastatic brain cancers, primarily via regulation of the estrogen receptor alpha (ERα) [25]. One study analyzed aromatase and ERα and ERβ expression in biopsy samples from 36 patients with grade I–IV astrocytoma [26]. Aromatase expression and estradiol levels in tumor tissues were significantly higher in grades III/IV astrocytoma, relative to grade II astrocytoma, and directly correlated with tumor grade [26]. In turn, patient survival negatively correlated with aromatase and estradiol concentration. In contrast, ERα abundance, a measure of estrogen bioactivity, negatively correlated with estradiol levels, tumor grade, and patient survival. Given the possible contribution of aromatase to brain cancer development, it was a logical extension to investigate the repositioning of AIs in GBM. Pre-clinical analysis of several aromatase inhibitors revealed that letrozole was the most effective against GBM cells and exhibited a satisfactory permeability through the BBB [27,28]. Based on these observations, ongoing research has focused on developing letrozole as an anti-GBM therapeutic and exploring its mechanism of action. 

In 1998, FDA approval was granted to Novartis for letrozole tablets (Femara^®^) as a second-line hormone based-chemotherapy for the treatment of advanced breast cancer in postmenopausal women with hormone receptor positive or unknown breast cancer after primary anti-estrogen therapy [29]. Letrozole also has an extensive history as an off label for the treatment of infertility. A phase 0–1 study or “Window-of-Opportunity trial” (NCT03122197) was initiated in 2017 to explore use of letrozole for treatment of recurrent gliomas. The study’s aim is to assess pharmacokinetic and pharmacodynamic properties of letrozole in recurrent glioma patients, including measurement of letrozole levels found in tumors of participants undergoing surgical resection to evaluate the ability of this molecule to penetrate the BBB. This trial is an ascending dose trial that will determine the maximal tolerated dose (MTD) of letrozole in the recurrent GBM setting and will provide valuable insights for future combination trials of letrozole with other targeted agents.

## 3. Repurposing an Immunosuppressive Agent That Inhibits an Enzyme Critical to Guanosine Nucleotide Synthesis

We have known for half a century the importance of purine nucleotides (e.g., ATP and GTP) in many cellular functions as a source of nucleotides for DNA and RNA synthesis, as a source of energy, as enzymatic cofactors in metabolic pathways, and as components in signal transduction. In addition to these established physiological roles, intracellular GTP concentrations are markedly elevated in many types of cancers, including in GBMs [30,31]. Until recently, it was thought to be a secondary, passive phenomenon. Recently, it was reported that elevated GTP synthesis increases cell anabolism and induces tumor malignancy through upregulation of inosine monophosphate dehydrogenase (IMPDH), the rate-limiting enzyme of the GTP nucleotide synthesis pathway [31,32] (Figure 1A).

Mycophenolic acid (MPA), and its pro-drug mycophenolate mofetil (MMF), are potent IMPDH inhibitors and an FDA approved (initial approval in 1995) potent immunosuppressant for treating autoimmune diseases and tissue transplanted patients. MMF (brand name: CellCept^®^) is used for a number of indications including preventing graft rejection. MMF is a member of the class of 2-benzofurans, consisting of a carboxylic ester resulting from the formal condensation between the carboxylic acid group of mycophenolic acid and the hydroxy group of 2-(morpholin-4-yl)ethanol (Figure 1B). MPA and MMF are also used as a second-line treatment for patients for whom corticosteroid treatment does not work adequately. Antitumor activity of MPA has been recognized since the late 1960s [35,36]. MPA was shown to inhibit cell proliferation in a broad range of systemic and CNS cancer cell lines, including neuroblastoma, lymphoma, pancreatic cancer, non-small cell lung adenocarcinoma, and colorectal cancer [37,38,39,40,41,42]. A phase I clinical trial was conducted with MMF for relapsed and refractory multiple myeloma in 2004 [43]. Doses ranged from 1 to 5 g/day, which were well tolerated. There was a significant correlation between the decrease in GTP levels of peripheral blood-derived mononuclear cells and the levels of MPA. This suggests the possibility of monitoring MMF activity in clinical practice, but the reason why peripheral blood GTP was only reduced in some patients is unclear. The slight in vivo anti-pancreatic cancer effect of MMF may be due to the fact that desmoplasia and stromal components, which have been proposed as a cause of drug resistance in pancreatic cancer, outnumbered the number of pancreatic tumor cells [44]. Although MMF has not been developed as a therapeutic agent for pancreatic ductal adenocarcinoma, these studies could serve as a benchmark for future phase 0 pharmacological trials with MMF in GBM and other tumors.

Recently, it has been reported that GBM and brain tumor initiating cells/glioma stem cells-like cells (GSCs) undergo an altered reprogramming of GTP metabolism [31,45,46]. Importantly, these studies showed that MMF treatment or genetic inhibition of IMPDH significantly decrease GBM growth in mouse models. Furthermore, MMF treatment sensitizes GBM cells to chemotherapy and radiotherapy [46,47]. 

However, a potential drawback of MMF or an IMPDH inhibitor for treating GBM is their potent immune suppressive effect, which may limit its use in an upfront setting. But this approach may find use for GBM associated edema treatment. Over 60% of GBM patients suffer from GBM-associated cerebral edema, a major cause of morbidity in GBM patients [48,49,50,51,52,53]. Cerebral edema causes symptoms such as headaches, cognitive and personality changes, seizures, delirium, and dysphagia. An accumulation of fluids in patients increases intracranial pressure, leading to ischemia, herniation, and ultimately death [54]. Furthermore, GBM-associated edema influences the clinical course and the prognosis of the disease [55,56]. Immunosuppressive corticosteroids have been the primary treatment for GBM-associated edema since the 1960s. While corticosteroids suppress the edema, the effect is temporary and accompanied by significant side effects (e.g., osteoporosis, myopathy, hyperglycemia) [57,58,59]. Importantly, recent studies show that corticosteroids reduce survival in a murine model [60] and human GBM patients [57,58,59]. Bevacizumab has an anti-edema effect; however, it does not extend patient survival [61,62,63]. Inflammation and neoangiogenesis, which destroy the integrity of the BBB causing fluid leakage, are two major causes of GBM-associated edema. Therefore, MPA/MMF treatment may have potential as a second-line treatment for GBM. This approach could both inhibit tumor growth and reduce cerebral edema for several reasons: (i) like corticosteroids, MPA and MMF are widely used as potent immunosuppressors for organ transplantation; (ii) they can suppress inflammation [64,65,66,67,68]; (iii) MPA treatment reduces VEGF secretion and neoangiogenesis in pancreatic cancer [69] and encapsulated peritoneal sclerosis [70]; (iv) cancer incidence is decreased in organ transplanted patients undergoing MMF treatment [71,72]; (v) as denoted above, the genetic and pharmacologic inhibition of IMPDH suppresses GBM growth in vivo [31,45,46]. 

Currently, there is a phase 0–1 Trial (NCT04477200) looking at the effects of MMF combined with radiation, based on the observation that MMF increases the efficacy of radiotherapy against GBM [47]. This is a dose-escalation study to determine the maximum tolerated dose of MMF when administered with radiation, in patients with recurrent GBM or recurrent gliosarcoma. 

## 4. Repositioning Two Agents to Target a Kinase Receptor Signaling Pathway 

Oncogenic kinase signaling is frequently activated in GBM. The human ‘kinome’ consists of over 500 kinases of which ~5% are affected by potential PanCancer driver events in the GBM PanCancer TCGA dataset [73,74,75,76]. Activating cancer driver events in the EGFR gene are the most frequently found kinase alterations in GBMs, yet early studies of EGFR inhibitor therapy did not yield clinical benefit [77,78]. However, tyrosine kinase inhibitors (TKIs) targeting EGFR have shown efficacy in previous studies on lung, colon, pancreatic, breast, and head and neck cancers [79]. Looking beyond EGFR targeting strategies for GBM treatment, genomic and expression analysis of GBMs has highlighted several onco-dependent kinase targets for potential therapeutic intervention, including the tyrosine kinases Met, FGFR, and Axl [80]. In a clinical trial of the small molecule kinase inhibitor cabozantinib as a monotherapy to target Met and Axl in recurrent GBM, there were some indications of clinical benefit in a small subset of patients [81]. This suggests an onco-dependency model for targeting the GBM kinome, wherein GBMs require signaling from Met, FGFR, Axl, and other kinases to sustain survival and metabolism, even as these kinases are intrinsically affected by oncogenic driver events. To counteract oncogenic kinase signaling in GBM, it is likely that kinase-targeting strategies will require combination approaches. One emerging kinase targeting strategy would combine the targeting of Axl with inhibition of the ribosomal protein S6 kinases (S6Ks), which is selectively cytotoxic for PTEN-deficient GBMs and leukemia cells [82]. Continued investigation of the mechanistic links between Axl and S6Ks will be needed to optimize strategies for counteracting oncogenic signaling in GBM. Liu et al. find that the S6K1 inhibitor, AD80, is selectively cytotoxic for PTEN-deficient cancer cells, while LY-2584702 is ineffective as a single agent. AD80 avoids S6K1 priming and co-targets Tyro-3, Axl, and Mer or TAM family of receptor tyrosine kinases. Combining LY-2584702 (Figure 1B) with the TAM kinase inhibitor BMS-777607 (Figure 1B) is selectively cytotoxic for PTEN-deficient cells (Figure 1A).

BMS-777607, an inhibitor of Met-related targets including c-Met and Axl, has entered into clinical study in Australia sponsored by Bristol-Myers Squibb [83]. The study (NCT00605618) evaluates dose and preliminary safety and efficacy of BMS-777607 in subjects with advanced or metastatic solid tumors. While GBM is not part of the study, findings can be used to support an IND to initiate human studies and optimize design of proposed clinical trials. There are five active clinical trials testing the Axl inhibitor BGB324 (bemcentinib), including a phase 1 trial in GBM (NCT03965494). The Met inhibitor PLB1001 (also known as AP-101 or bozitinib) completed a phase 1 clinical trial with the observation of a partial response in 2/18 patients with high grade gliomas characterized by Met-alterations [84]. Additionally, PLB1001 is being investigated (NCT03175224) in glioma and other advanced tumors. 

## 5. Repurposing a Derivative of an Antidepressant Found to Inhibit Cancer Cell Migration

Imipramine Blue (IB) is an organic triphenylmethane dye modeled after Gentian Violet, which has shown anti-cancer attributes (Figure 1B) [85]. IB is thought to have an effect on cell migration due to its ability to inhibit NADPH oxidase and limit actin fiber formation (Figure 1A) [86]. Reactive oxygen (superoxide) stimulates the polymerization of actin into filamins required for migration [86]. IB has demonstrated activity against other invasive tumors, including head and neck cancer, triple negative breast cancer, acute myeloid leukemia, and chronic myelogenous leukemia [87,88,89,90]. Since IB is derived from Imipramine, a tricyclic antidepressant known to cross the BBB, it is logical that this drug could overcome treatment challenges for GBMs. In addition, its anti-cell migration feature makes it a good drug candidate to halt metastasis and allow for more efficient chemotherapeutic intervention [86]. A study conducted in 2012 showed promising results in a rat model with glioma cell lines [86]. In this study, the growth and infiltration of tumor cells into surrounding healthy tissue was reduced, allowing the chemotherapy agent doxorubicin to more effectively target cancer cells in a localized area [86]. 

Next steps for IB include testing its pharmacokinetics and toxicology in preparation for pre-IND studies for eventual human clinical trial. Prior FDA approval and use of Imipramine for over 50 years to treat depression and anxiety should provide significant information to guide preclinical safety studies and phase 1 clinical trial design for IB. Data for Imipramine Blue’s approval and its history of use in patients should facilitate the IND process required to initiate first in human studies to treat metastatic cancers. This will reduce both time to commercialization and development costs if IB proves to be an effective treatment of GBM. 

## 6. Repurposing a Drug for Macular Degeneration

Verteporfin (VP) is a small benzoporphyrin derived molecule originally used to treat macular degeneration (Figure 1B). The FDA approved its use in 2002 for ocular treatment. Specifically, VP is used as a photosensitizer in photodynamic therapy to generate localized free radicals that reduce formation of blood vessels [91]. VP by itself can also act as an inhibitor of the Yes-associated protein (YAP) and PDZ-binding motif (TAZ) family transcriptional coactivators [34]. YAP and TAZ play important roles in the Hippo (Hpo) and receptor tyrosine kinase (RTK) signaling pathways and are proven onco-proteins in several tumor types, including GBMs (Figure 1A) [92,93,94,95]. Previous studies have shown that VP has cytotoxic effects in GBM cells and cells of other cancer types including colon, endometrial, cervical, breast, pancreatic, and melanoma cancers [96,97,98,99,100,101,102,103]. Porphyrins and their precursors have been the subject of intensive development for brain tumor therapy and intraoperative brain tumor imaging because GBM and glioma cells readily absorb these molecules in vitro and in vivo in animal models and patients [104,105,106].

A phase 0 clinical trial of Visudyne^®^, the brand name for an FDA-approved liposomal formulation of VP, confirmed that VP is readily absorbed into tumor cells in GBM patients [107]. A phase 1–2 trial is in progress, to test VP dose escalation and efficacy in recurrent GBM (NCT04590664) with the goal of expanding testing to primary GBM. The study will look at the incidence of adverse events and provide an initial assessment of efficacy for this new formulation and route of administration of VP for treatment of brain tumors. To optimize CNS absorption for GBM and other brain tumors, studies successfully tested reformulated versions of VP paired with nanoparticles or poly(ethylene glycol)-poly(β-amino ester)-poly(ethylene glycol) (PEG-PBAE-PEG) micelles in cell culture and mouse xenograft GBM models [108,109]. Other ongoing investigations of the therapeutic potential of VP for GBM are aimed at determining whether it can synergize with other therapeutics, such as the standard of care chemotherapeutic temozolomide (brand name: Temodar), as well as other therapeutic modalities, i.e., radiotherapy. 

## 7. Targeting a Phospholipid with a Saposin C Embedded Nanoparticle

Above we have offered several case studies of therapeutics being repositioned or repurposed for GBM. In this section, we highlight a novel approach that combines use of a small lysosomal and multifunctional glycoprotein, saposin C (SapC) [110], embedded in the membrane of a microvesicle rich in negatively charged phospholipids (Figure 2).

SapC functions to activate the lysosomal metabolic enzyme β-glucosidase to hydrolyze glucosylceramide by cleaving the β-glucosidic linkage between glucose and ceramide in the presence of acidic phospholipids at pH 4–5 [112]. Deficiency of acid β-glucosides in humans leads to the lysosomal storage disorder Gaucher disease, which is caused by the accumulation of sphingolipids and saposins in Gaucher cells [110,113]. This abnormal build-up in low pH lysosomes is thought to be toxic to monocytes and macrophages. The microenvironment surrounding cancer cells and tissues appears acidic under hypoxic stress [114]. 

Neoplastic cells are predicted to be sensitive to cytotoxicity of the saposin-fat complexes. As a membrane-associated protein, SapC can tightly bind the negatively charged phospholipids (DOPS) to form a stable and pharmacologic active nanovesicle, SapC-DOPS [115,116]. This “nanodrug” selectively targets phosphatidylserine, a surface lipid biomarker on tumor cells and vessels [117,118]. Tumor-specific cytotoxicity of SapC-DOPS on a variety of cancer types leads to apoptotic and lysosomal cell death, thus inhibiting tumor growth and improving survival of tumor-bearing animals [119,120]. SapC-DOPS has been previously studied in pancreatic, lung, pediatric, and other brain tumors [116]. As for suggesting its use in the GBM space, SapC-DOPS penetrates the BBB and BTB to regress brain tumors in mice [116,121]. In addition, SapC-DOPS technology may potentially find use as a carrier of imaging agents to a tumor [114,122,123].

Based on strong evidence of preclinical studies, Bexion Pharmaceuticals licensed the SapC-DOPS anti-cancer technology from Cincinnati Children’s Hospital Medical Center in 2006. The SapC-DOPS nanodrug (BXQ-350; B = Bexion, X = Xiaoyang, and Q = Qi) completed phase 1 trials in both adult (NCT02859857) and pediatric (NCT03967093) populations, which established the safe dose for treatment of recurrent high-grade gliomas and generated pharmacokinetic and safety profiles. In addition, phase 1 studies provide a preliminary assessment of anti-tumor activity of BXQ-350 administered at the MTD, or the maximum dose level proposed if the MTD is not reached. This study also includes adult participants with advanced solid tumors and pediatric and young adult participants with relapse solid tumors which contributes to the establishment of a strong human safety profile. Phase 2 studies are currently being planned. Bexion has also received orphan drug and rare pediatric disease designations from the FDA to support the commercialization of BXQ-350 for the treatment of adult and pediatric malignant gliomas, which could expedite its regulatory approval for this indication [124].

## 8. Conclusions

The drug discovery pathway currently emphasizes monotherapy of a drug designed for a particular target or pathway. Human safety and toxicities of new drugs are unknown until trialed, which poses a significant time delay for viable treatment options. In contrast, bench research and clinical trials with already established drugs can be explored more quickly, possibly offering a timely treatment to patients diagnosed with GBMs. With such a small window of survivorship following GBM diagnosis, it is important that drug exploration in this field be explored quickly in order to provide as many possible treatment options as possible to patients.

The cytoxicity of repurposed-repositioned drugs on cancer cells can be assessed based on the drug’s target and role. Letrozole is being repositioned to the neuro-oncology space from its common role of treating breast cancer because it poses potential in inhibiting cell migration and proliferation and reducing tumor growth in GBMs. It accomplishes minimizing these common tumor characteristics by inhibiting estrogen synthase and reducing the concentration of estrogen in glial cells. In addition to letrozole, CellCept has been repurposed due to its expected efficacy in decreasing cell proliferation, anabolism, and tumor malignancy. By inhibiting IMDPH, CellCept has the ability to decrease GTP synthesis. LY-2584702 and BMS-777607 are able to be repositioned to treat GBMs together because of their ability to block kinase receptor signaling pathways and the S6K1 enzyme. Inhibiting these provides potential for decreasing apoptosis resistance, mitogenesis, and metabolic events in cancer cells. Imipramine Blue is being repurposed with hopes to limit cell migration and decrease inhibition of transcription factors known to aid cell survival. By inhibiting NADPH oxidase and reducing the amount of reactive oxygen species within cells, IB may offer these promising anti-cancer attributes in the GBM space. Verteporfin targets the interaction between YAP/TAZ and TEAD in order to regulate the Hippo pathway. As a result, VP assists in cell regulation including apoptosis and cell proliferation control. Although there is no specific pathway in which SapC-DOPS inhibits or promotes to offer anti-cancer benefits, the nanovesicle is proposed to offer inhibition of tumor growth through apoptotic and lysosomal death. 

Previously established drugs are beginning to enter the poly-pharmacology research space with an end goal to treat GBMs and halt their inevitable recurrence. However, there are limited options available aside from the standard of care, as these drugs are still undergoing extensive research and clinical trials [13,14]. Yadavalli et al. and Tan et al. have reviewed many of these drugs and shared their progress and potential in offering effective treatment [13,14]. Repurposing and repositioning drugs shows potential promise in identifying novel treatment options for GBMs or laying groundwork for future pharmaceutical interventions that will assist in identifying means for tumor suppression. Above we have drawn attention to several molecules that have been recently repositioned or repurposed for use as anti-GBM therapeutics at our institutions and others. These molecules target different aspects of the cancer cell and they are being translated in different ways; some of which have shown promising results in translational research and have since moved on the clinical studies. Rather than proposing a single, monotherapy solution to primary brain tumors, we imply the drugs we discuss may be applicable in other brain tumor settings, such as in GBM recurrence or secondary tumor settings. In addition, these drugs have potential to be explored concomitantly with other known cancer therapy modalities such as chemotherapy, radiation, or TTFs. Such combinations may offer benefit to patients diagnosed with GBMs. In the future, it is possible that these drugs could be considered for therapeutic substitutions to patients who experience allergies or adverse reactions to the current standard of care drugs, should they show efficacy in clinical trials. 

## Figures and Tables

**Figure 1 pharmaceuticals-14-00099-f001:**
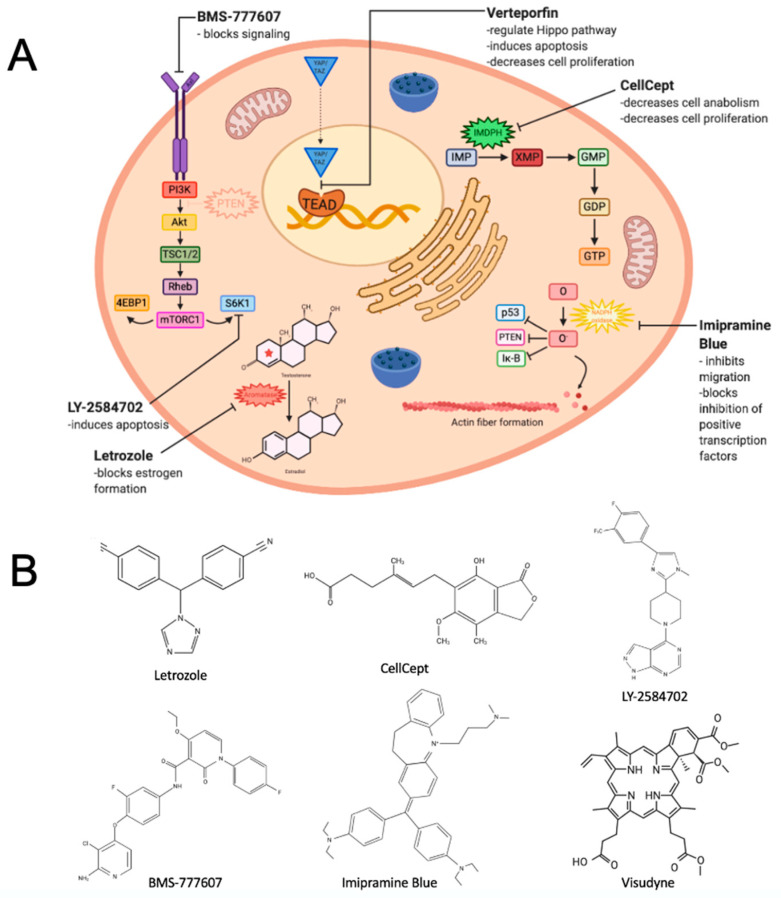
Panel (**A**): Mechanisms of drug inhibition on cellular pathways. (Letrozole). Aromatase catalyzes the conversion of androgens to estrogens by aromatization of the A-ring. A-ring structure of androgens is noted by the red star. (CellCept) GTP synthesis increases cell anabolism through upregulation of inosine monophosphate dehydrogenase, IMPDH, the rate-limiting enzyme of the GTP nucleotide synthesis pathway [31,32]. IMPDH is inhibited by CellCept. (BMS-777607 and LY-2584702) Shown is the canonical oncogenic kinase signaling pathway downstream of TAM receptor, Axl. PTEN deficiency is depicted by transparency. (Imipramine Blue) Imipramine Blue, shown here in schematic, is thought to have an effect on cell migration due to its ability to inhibit NADPH oxidase, which catalyzes electron transfer to oxygen from NADPH, thereby limiting actin fiber formation. Importantly, O_2_^−^ inhibits transcription factors critical to cell survival, including PTEN, Iκ-B, p53 [33]. (Verteporfin) Verteporfin can act as an inhibitor of the Yes-associated protein (YAP) and PDZ-binding motif (TAZ) family transcriptional coactivators [34], which have roles in the Hippo and RTK signaling pathways. Verteporfin disrupts the interaction between YAP and TEAD transcription factors, which regulate the Hippo signaling pathway. Panel (**B**): Chemical structure of the molecules discussed above.

**Figure 2 pharmaceuticals-14-00099-f002:**
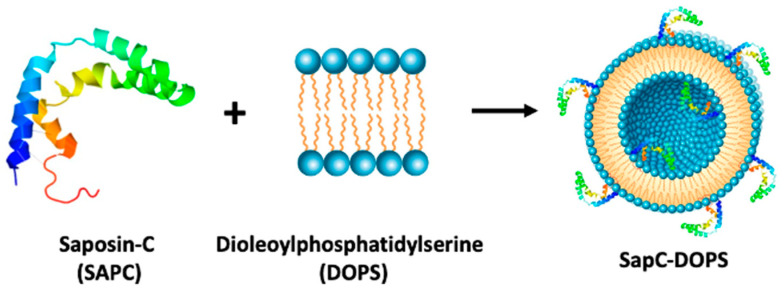
Makeup of SapC-DOPS. Nanodrug SapC-DOPS is composed of a small lysosomal and multifunctional glycoprotein, SapC (Saposin C) [110], embedded in the membrane of a microvesicle rich in negatively charged phospholipids, dioleoylphosphatidylserine or 1,2-dioleoyl-sn-glycero-3-phosphoserine (DOPS). Figure 2 is adapted from [111].

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
