# Peer review of "Reuse of Molecules for Glioblastoma Therapy"

_pharmaceuticals, 2021, doi:10.3390/ph14020099_

Round 1

Reviewer 1 Report

This work is a revised version of a previous manuscript on the repurposing of some drugs for glioblastoma (GBM) therapy. 

The work is well written and organized. This time the presence of some additional paragraphs and figure 1 are useful to get the main message of this work. The authors have really improved this paper.

Reviewer 2 Report

Glioblastoma (GBM) is the most malignant of primary brain tumors and one of the leading causes of cancer-related deaths in the United States.  Treatment methods including surgery, radiation therapy, and chemotherapy are available for the cure. However, the median survival rate of patients remains at 14.6 months after diagnosis of GBM, made worse by the high rate of relapse after surgery. Numerous factors, such as resistance to conventional chemoradiation and differential response rates of heterogeneous cancer cell populations within tumors limit GBM therapies. The most commonly used adjuvant chemotherapeutic drug for GBM is temozolomide (TMZ),  a prodrug of the alkylating agent 5-(3 methyltriazen-1-yl) imidazole-4-carboximide (MTIC), that disrupts DNA replication and causes programmed cell death (apoptosis) in rapidly dividing cells.  Even though TMZ is well tolerated by patients, it is however associated with side effects. In the present review, the authors aimed to present selected molecules that can be considered as a treatment option for GBM. The manuscript is reviewed well and I recommend to the editor  to accept in the present from.

This manuscript is a resubmission of an earlier submission. The following is a list of the peer review reports and author responses from that submission.

Round 1

Reviewer 1 Report

This manuscript deals with the selection of drugs that can be repurposed or repositioned for glioblastoma (GBM) treatment. The use of these drugs for GBM is proposed based on their use in specific clinical studies.

The manuscript is clearly written and it considers some drugs that have not been considered in other recent (and more extensive and detailed) published review on this topic.

On one hand this work gives some interesting suggestions on the topic and at the same time it is less detailed and comprehensive than others present in the literature and referenced by the  authors.

The repurposing and repositioning of aromatase inhibitors, mycophenolic acid (MPA), and its pro-drug mycophenolate mofetil (MMF), some kinase inhibitors, imipramine blue, verteporfin and saposin C for the treatment of glioblastoma are of interest but almost the large majority of anti-tumor drug. Indeed, all the examples selected have been used in tumors other than glioblastomas too.

I would consider this work with a low priority rate for publication.

Reviewer 2 Report

The introduction should be updated according to molecular classification.
The structure of the article should be improved, point 2 discussion I think it not a discussion but a description.
A big picture should be shown at the end. A figure with main pathways which these inhibitors act.
The conclusion is so short and did not improve the stat of the art neither showed the possible future. This article needs to be more critical and more interventional with ideas for the future.
Panel A of each picture I think is not necessary. The work is usually done for a team.